# Diffusion-synthesized Chest X-rays improve fairness and diagnostic performance

**Dawood Rehman**[1], **Huan-Yu Chen**[2], **Chi-Chun Lee**[1,2], **Natalia Vilor-Tejedor**[3,4], **Shreya Upadhyay**[2], **Po-Chih Kuo**[1,5]*

**1** Ph.D. Program in Biomedical Artificial Intelligence, National Tsing Hua University, Hsinchu, Taiwan, **2** Department of Electrical Engineering, National Tsing Hua University, Hsinchu, Taiwan, **3** Institute for Risk Assessment Sciences, Department of Veterinary Medicine, Utrecht University, Utrecht, Netherlands **4** BarcelonaBeta Brain Research Center, Pasqual Maragall Foundation, Barcelona, Spain, **5** Department of Computer Science, National Tsing Hua University, Hsinchu, Taiwan

* kuopc@cs.nthu.edu.tw

## Abstract

Deep learning models have been widely applied to chest X-ray (CXR) disease classification and diagnosis; however, challenges such as data scarcity and shortcut learning often lead to biased model behavior. This study addresses fairness-related concerns in conventional deep learning models trained on CXR data and proposes mitigating demographic disparities through image synthesis. We fine-tune a pre-trained stable diffusion model using Low-Rank Adaptation (LoRA) and a CLIP tokenizer, incorporating low-rank constraints into key attention layers while preserving the original architecture. This enables the generation of high-quality, realistic CXR images with reduced parameter complexity. Experimental results demonstrate that models trained with our synthetic data achieve improved classification performance and exhibit significantly reduced disparities across demographic groups. Furthermore, the proposed models show increased attention to disease-relevant regions and diminished reliance on spurious shortcuts. These findings highlight the potential of generative AI in enhancing fairness in medical imaging workflows, particularly when combined with efficient and adaptable fine-tuning strategies.

## Author summary

Medical image analysis is one area where artificial intelligence has shown a lot of promise, but its effectiveness is mostly dependent on huge, well-balanced datasets. Many of the chest X-ray collections that are now available are small and skewed towards particular patient populations, which may result in inaccurate or unjust projections. In this work, we develop realistic, text-guided synthetic chest X-rays using a generative AI model based on diffusion techniques. We discover that models trained on synthetic data outperform models trained on real data. The most accurate and equitable findings are obtained when real and

**Data availability statement:** All codes, the synthetic dataset, and the model weights discussed in this article are publicly available for experimentation and reproducibility via this link: https://github.com/dawoodrehman44/Stable-Diffusion-with-LoRA.

**Funding:** This work was supported by the Ministry of Science and Technology, Taiwan (114-2628-E-007-008-MY4 to PCK) and U of I-UAAT Joint Research Project (113M7054 to PCK). The funders had no role in study design, data collection and analysis, decision to publish, or preparation of the manuscript.

**Competing interests:** The authors have declared that no competing interests exist.

synthetic data are combined. According to our research, synthetic imagery may be a useful tool for enhancing medical AI performance, improving model focus on disease regions, reducing bias, and improving the fairness and reliability of diagnostic systems for various patient demographics.

## Introduction

Over the past decade, deep learning has revolutionized medical imaging, particularly in disease detection and diagnosis using chest X-rays (CXRs). Models such as Convolutional Neural Networks (CNNs), Vision Transformers (ViTs), and hybrid architectures have achieved performance comparable to expert radiologists in detecting conditions like pneumonia, tuberculosis, and COVID-19. These models have the potential to reduce clinician workload, accelerate diagnosis, and improve patient outcomes.

Despite these successes, AI models in medical imaging are vulnerable to fairness and generalizability issues. Training datasets often fail to adequately represent the diversity of patient populations, imaging devices, and acquisition protocols. Consequently, deep learning models can rely on spurious correlations, or "shortcuts," which may include demographic cues such as age, gender, race, or scanner-specific artifacts, rather than disease-specific features. While these shortcuts can boost apparent performance on the training set, they often lead to significant disparities in predictions for underrepresented or out-of-distribution groups. This poses ethical concerns, as misdiagnosis or reduced performance for certain populations can exacerbate healthcare inequalities and compromise patient safety.

The problem of fairness in medical AI has been well-documented. Studies have demonstrated that CNNs trained on large datasets can exhibit performance gaps across racial, gender, and socioeconomic groups [1,2]. For instance, models trained on chest X-rays may systematically underperform on images from Black or Asian patients compared to White patients or may classify disease differently in males versus females. Even seemingly subtle image artifacts, such as tokens, hospital labels, or device-specific noise, can unintentionally influence model predictions [3,4]. These findings underscore the importance of not only high accuracy but also equitable performance across diverse patient populations, particularly as AI tools are increasingly integrated into clinical decision-making.

Addressing fairness and bias in medical imaging has been the focus of numerous studies. Traditional approaches have relied on data-centric methods, including data augmentation, resampling, and synthetic data generation. Techniques such as image rotation, scaling, noise injection, and adversarial U-Nets can increase the diversity of the training set and reduce overfitting [5,6]. Transfer learning, where models pretrained on large datasets are fine-tuned on smaller, domain-specific datasets, is also widely used to improve model generalization across different hospitals, populations, and imaging devices.

However, these methods have limitations. Standard augmentation techniques often fail to generate truly diverse images, especially when demographic subgroups are underrepresented. They also do not address biases inherent in the labels, which may reflect historical inequities or annotation inconsistencies. Moreover, simple augmentation cannot fully prevent models from relying on spurious correlations encoded in the training data. These limitations have motivated the use of generative AI methods, particularly diffusion models, for fairness-aware data generation.

Diffusion models [7,8] are a class of generative models that iteratively refine noisy inputs to produce high-fidelity synthetic images. In medical imaging, diffusion models have been applied for data augmentation, synthetic dataset creation, anomaly detection, and complex imaging tasks. Their ability to preserve fine-grained anatomical and pathological details makes them particularly suitable for generating synthetic CXRs that reflect diverse patient populations. Studies such as [9] have demonstrated that diffusion-based synthetic data can improve model generalization and reduce bias in disease classification tasks.

Recent works have explored various applications of diffusion models in chest radiography. Latent diffusion models have been fine-tuned to improve image quality and disease representation [10], while frameworks such as Xreal, DINO-Diffusion, and Chest-Diffusion focus on anatomical control, self-supervised learning, and report-guided generation [11,12]. Other studies, including BS-Diff and DDPM-based synthesizers, target specific tasks like bone suppression and pneumonia image generation [13,14]. Additionally, models such as FairDiffusion and PRISM explicitly address fairness and spurious correlations, demonstrating the potential of generative approaches for bias mitigation [15,16].

Despite these advances, most prior studies focus on either single-disease scenarios or demographic fairness in isolation. Few approaches simultaneously improve model focus on disease-relevant features while mitigating bias across multiple demographic dimensions, such as race, age, and gender. This gap motivates the proposed study, which leverages diffusion-based synthetic CXRs to enhance both disease-specific learning and fairness in model predictions across diverse patient populations.

The main objectives of our study are as follows:

- Use diffusion models to create synthetic chest radiographs that improve demographic representation in training data, allowing greater control over encoded information.

- Utilize pre-trained models with billions of parameters and employ low-rank adaptation (LoRA) and prompt engineering to avoid training the entire UNet architecture from scratch.

- Enhance CNN performance in disease classification by directing model attention toward disease-relevant regions instead of demographic shortcuts.

- Mitigate disparities in model performance across demographic groups, specifically focusing on race (e.g., White, Black, Asian), gender (male and female), and age groups.

## Materials and methods

### Conversion of Tabular data to text

Text-guided diffusion models such as Stable Diffusion [17] require image–text pairs for effective training. However, the CheXpert dataset provides only structured tabular annotations comprising binary disease labels, demographic attributes, and support device indicators. To bridge this gap, we adopt a template-based conversion strategy following [16] that transforms structured clinical data into natural language descriptions. Each patient record is systematically converted into a descriptive caption conditioned on its diagnostic labels and metadata. For instance, a chest X-ray labeled with *pneumonia* and *enlarged cardiomediastinum* generates the caption: *"Chest X-ray of a patient showing pneumonia and enlarged cardiomediastinum."* When demographic information is available, it is incorporated to enhance specificity, yielding captions

such as: *"Chest X-ray of a 55-year-old male patient showing pneumonia and support devices."* This templating approach ensures that each training image is paired with a semantically coherent and clinically relevant prompt, establishing the necessary text-image alignment for conditioning the diffusion process during model training.

### Latent diffusion model with domain-specific LoRA Adaptation

We employ Stable Diffusion v1.5 [17] as our foundational architecture due to its demonstrated capacity for high-fidelity image synthesis and computational efficiency through latent space operations. Unlike pixel-space diffusion models that operate directly on high-dimensional image data, latent diffusion compresses images into a lower-dimensional latent representation, significantly reducing memory requirements and training time while preserving semantically relevant features and properties, particularly advantageous for medical imaging, where computational resources are often constrained and subtle anatomical details must be maintained.

The architecture comprises three principal components: (i) a Variational Autoencoder (VAE) that encodes chest X-rays from pixel space (512 × 512) into a compressed latent representation (64 × 64 × 4) with 8 spatial down sampling factors, and subsequently decodes the denoised latent back to pixel space; (ii) a UNet backbone operating in latent space to iteratively predict and remove noise across diffusion timesteps; and (iii) a frozen CLIP text encoder that maps clinical text descriptions into semantic embeddings, enabling precise text-to-image conditioning through cross-attention mechanisms within the UNet.

### Diffusion process formulation

The forward diffusion process systematically corrupts a latent representation $x_0$ (encoded from the original chest X-ray) by progressively adding Gaussian noise over $T = 1000$ time steps, ultimately transforming structured medical images into approximately isotropic Gaussian noise. We employ a linear variance schedule with $\beta_1 = 0.0001$ and $\beta_T = 0.02$, which has been empirically validated for medical image synthesis tasks as it provides stable training dynamics and preserves fine-grained pathological features during the noising trajectory. The reverse diffusion process learns to denoise corrupted latents by modelling the posterior distribution:

$$p_\theta(x_{t-1} \mid x_t) = \mathcal{N}\left(x_{t-1}; \mu_\theta(x_t, t), \Sigma_\theta(x_t, t)\right),$$

$$(1)$$

where the predicted mean $\mu_\theta$ is computed via the noise predictor $\epsilon_\theta$, parameterized by the UNet:

$$\mu_\theta(x_t, t) = \frac{1}{\sqrt{\alpha_t}}\left(x_t - \frac{1 - \alpha_t}{\sqrt{1 - \bar{\alpha}_t}}\,\epsilon_\theta(x_t, t, c)\right),$$

$$(2)$$

where $c$ denotes the text-conditioning embedding derived from the clinical prompts. The training objective minimizes the simplified denoising loss, encouraging the model to accurately predict the noise component $\epsilon \sim \mathcal{N}(0, \mathbf{I})$ added at timestep $t$, conditioned on the clinical text.

### Parameter-efficient adaptation via Low-Rank Decomposition

Chest X-rays exhibit domain-specific characteristics that distinguish them from natural images: (i) constrained anatomical variability with strict spatial structure governed by skeletal and organ geometry; (ii) subtle pathological manifestations requiring precise localization and appearance modeling; and (iii) specialized clinical terminology in radiological reports necessitating refined text-image semantic alignment.

Full fine-tuning of Stable Diffusion's 860M parameters on limited medical datasets risks overfitting and catastrophic forgetting of the pre-trained model's generative capabilities. To address these challenges, we adopt Low-Rank Adaptation

(LoRA) [18], a parameter-efficient fine-tuning technique that introduces trainable low-rank matrices into the model's attention layers while keeping the majority of pre-trained weights frozen.

Specifically, for a pre-trained weight matrix $W \in \mathbb{R}^{d \times k}$ in an attention projection layer, LoRA decomposes the weight update as a low-rank factorization:

$$W' = W + \Delta W = W + BA, \quad A \in \mathbb{R}^{r \times k}, B \in \mathbb{R}^{d \times r}, r \ll \min(d, k), \tag{3}$$

where $r$ is the rank of the decomposition, and only the low-rank matrices $A$ and $B$ are trainable, while the original weights $W$ remain frozen. This drastically reduces the number of trainable parameters from 860M to approximately 2.4M (a 99.7% reduction), enabling efficient fine-tuning on medical datasets while mitigating overfitting risks. We apply LoRA modules exclusively to the query (`to_q`), key (`to_k`), and value (`to_v`) projection matrices within both self-attention and cross-attention layers of the UNet. The `to_q` projections determine how each latent feature queries relevant spatial or semantic information, allowing the model to focus on diagnostically important regions. The `to_k` projections define the searchable keys that encode contextual relationships among visual and textual tokens, enabling accurate feature correspondence. The `to_v` projections modulate the actual information passed through the attention mechanism, refining visual detail and enhancing alignment between medical image structures and their clinical descriptions. In total, LoRA is injected into 48 attention projection layers across the UNet's encoder and decoder blocks, encompassing all 16 attention layers with three projections each.

## Hyperparameter selection for medical domain adaptation

The rank parameter $r$ governs the expressiveness of the low-rank adaptation: lower values enforce stronger regularization and parameter efficiency, while higher values provide increased model capacity at the cost of potential overfitting. The scaling factor $\alpha$ controls the magnitude of the low-rank updates, with the effective adaptation strength proportional to $\alpha/r$. For our experiments, we set $r = 4$ and $\alpha = 16$, yielding an effective scaling of 4.0. This configuration balances two competing objectives: (i) maintaining sufficient representational capacity to learn domain-specific radiological features and clinical text alignments present in chest X-ray data, and (ii) preserving the regularization necessary to prevent overfitting on the relatively limited medical training set compared to natural image datasets used for pre-training. The choice of $r = 4$ represents 0.28% of the typical attention dimension (1024 or 1280 in Stable Diffusion v1.5), ensuring minimal parameter overhead while empirically demonstrating adequate adaptation capability across diverse pathological conditions and anatomical variations. This parameterization reduces GPU memory requirements from 24GB to approximately 8GB during training and decreases fine-tuning time by approximately 75% compared to full parameter updates, making the approach feasible for clinical research settings with limited computational infrastructure.

## Architecture overview and information flow

Fig 1 illustrates the complete pipeline of our LoRA-adapted latent diffusion model for chest X-ray synthesis. During training, an original chest X-ray image $A^{orig}$ is encoded by the frozen VAE encoder into a latent representation $O_l \in \mathbb{R}^{64 \times 64 \times 4}$. This latent undergoes the forward diffusion process, where Gaussian noise is progressively added over time steps $K_0 \rightarrow K_T$ until the latent becomes nearly indistinguishable from pure noise. Concurrently, the corresponding clinical text prompt $P_{orig}$ (or its edited variant $P_{edit}$) is processed by the frozen CLIP text encoder, producing a conditioning embedding $O_P \in \mathbb{R}^{77 \times 768}$ that encapsulates the semantic content of diagnostic findings and clinical context. During the reverse diffusion process ($K_T \rightarrow K_0$), the UNet iteratively predicts and subtracts noise from the corrupted latent. At each denoising step, the text embedding $O_P$ modulates the UNet's cross-attention layers, guiding the generation process to align with the clinical prompt, for instance, ensuring anatomically correct placement of pathologies or medical devices as specified in the text. The LoRA modules, integrated into the attention projections and depicted with flame icons in the Fig 1, enable

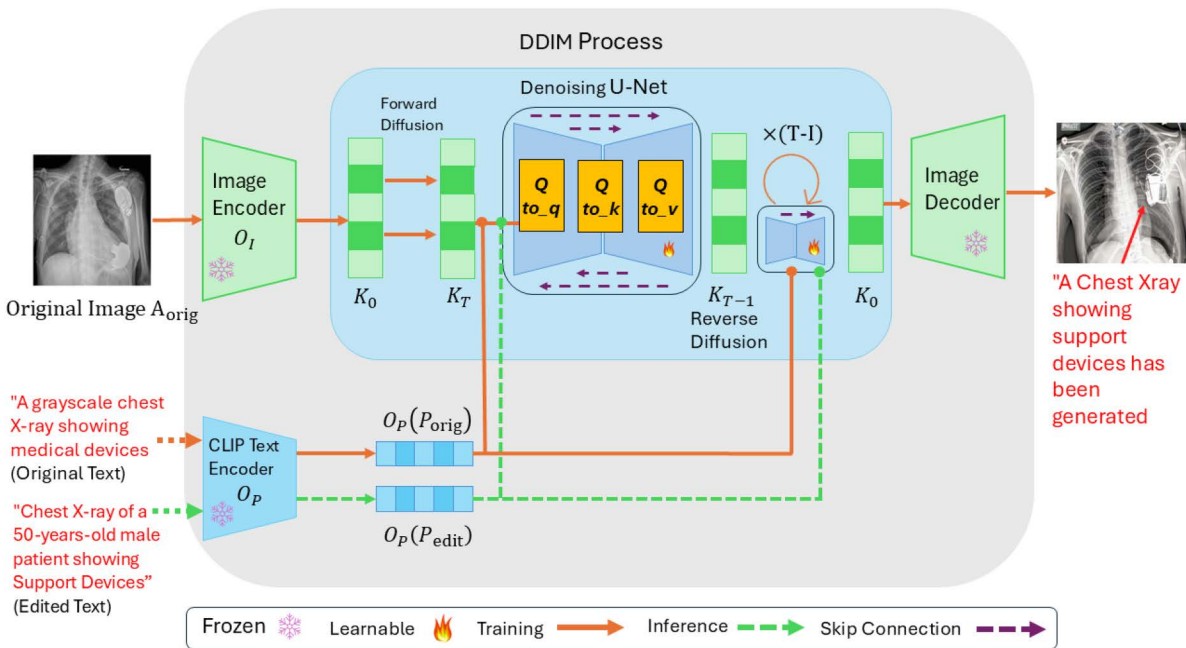

**Fig 1. Proposed Fine-Tuned Stable Diffusion Model with LoRA.** During the inference, the edited text used a prompt for generating high-quality 512 × 512 chest X-rays.

fine-grained control over this text-to-image alignment by learning medical domain-specific query, key, and value transformations. Skip connections within the UNet architecture (illustrated by lateral pathways) preserve multi-scale spatial information and anatomical structure throughout the denoising trajectory. Finally, the fully denoised latent $K_0$ is passed through the frozen VAE decoder to reconstruct a high-resolution chest X-ray in pixel space, semantically consistent with the guiding clinical text. The Fig 1 explicitly distinguishes frozen components (snowflake), trainable LoRA parameters (flame), inference pathways (orange arrows), and text conditioning flows (green dashed arrows), providing visual clarity on the training dynamics and information propagation through the network.

## Training and dataset integration

Training was performed using the CheXpert dataset's structured labels converted into descriptive captions. For each disease category, multiple synthetic images were generated by conditioning on captions (e.g., *"Chest X-ray of a patient showing pneumonia"*). Demographic attributes (age, sex) and support device indicators were included in prompts for more diverse image synthesis. Generated images were saved in CheXpert-compatible directory structures with accompanying CSV metadata. This allowed seamless integration with existing pipelines for medical image analysis.

## Experiments and implementation details

### Datasets

For this study, we utilized four datasets: CheXpert [19], MIMIC-CXR [20], ChestX-ray14 [21], and a synthetic dataset. CheXpert is an open-source dataset comprising approximately 224,316 chest radiographs (CXRs) from 65,240 patients, including disease labels, age, sex, race, and ethnicity information, collected from radiographic examinations at Stanford Hospital between 2002 and 2017. From this dataset, we constructed a balanced subset of 130,000 images based on disease labels and demographic groups to ensure fair representation across classes. MIMIC-CXR is another large

publicly available dataset containing around 370,000 radiographs collected at the Beth Israel Deaconess Medical Center in Boston, USA. To maintain consistency with CheXpert, we selected a subset of approximately 130,000 images. We also included the ChestX-ray14 dataset, which consists of frontal-view CXRs with 14 disease labels and has been widely used in previous studies due to its relevance to CheXpert. Finally, we developed a synthetic dataset designed to mimic the CheXpert style with respect to both disease and demographic distributions. The synthetic dataset contains approximately 84,000 images, with each disease label represented by an equal number of 6,000 samples, and around 10% of the dataset annotated with demographic information. For all four datasets, we applied the same data partitioning strategy, allocating 70% of the samples for training, 15% for validation, and 15% for testing, ensuring consistency across real and synthetic datasets [19–21].

### Data pre-processing

During data pre-processing, uncertain labels were excluded across all datasets to avoid ambiguity, and lateral-view CXRs were removed due to their relatively small representation compared to frontal views. Additionally, all input images were resized to 320 × 320 pixels for model training, with normalization applied using mean values of [0.485, 0.456, 0.406] and standard deviations of [0.229, 0.224, 0.225]. To improve generalization, we applied mild data augmentation techniques such as brightness adjustment, zooming, and horizontal flipping. It is important to note that the diffusion model generates images at a resolution of 520 × 520 pixels; however, for consistency, all models were trained with 320 × 320 input sizes for both real and synthetic datasets.

### Experiment Setup

In general, experiments have been divided into two parts. Part (a) is related to the proposed stable diffusion model by generating images with different prompts and then evaluating their quality. In addition, we performed an ablation study for the target modules and LoRA rank to investigate the impact of the steps of the LoRA model. In part (b), we use both the synthetic and the original datasets for experiments related to the models' classification, focus-observation performance, and fairness testing. The model will then be tested on real images.

### CXR-CLIP

We fine-tuned the CXR-CLIP model for multilabel disease classification. It is a vision-language pre-trained model that aligns chest X-ray images with their corresponding textual reports in a joint embedding space [22]. For our task, we adopted its ResNet-based image encoder variant, $f_\theta : \mathbb{R}^{H \times W \times 3} \to \mathbb{R}^d$, which maps an input image $x$ to a latent feature representation $f_\theta(x)$. We then append a linear classification head $g_\phi : \mathbb{R}^d \to \mathbb{R}^C$, where $C$ is the number of disease classes. Fine-tuning was performed end-to-end, with the image encoder initialized from pre-trained weights and the classifier trained from scratch. We chose CXR-CLIP over conventional CNN-based classifiers (e.g., DenseNet, ResNet) because of its ability to incorporate semantic medical knowledge from radiology reports. We also fine-tuned a variant of CXR-CLIP called Swin Transformer (Swin Tiny), which is a hierarchical vision transformer model designed to improve efficiency and scalability in image recognition tasks. Swin Tiny introduces a shifted window attention mechanism that computes self-attention within local windows but shifts these windows between layers to enable cross-window connections. It has been shown to achieve competitive accuracy on various vision benchmarks and is well-suited for medical imaging tasks. For all our experiments throughout the paper, we use these two models as our models with the proposed approach.

### Implementation details

The fine-tuned Stable Diffusion model was trained with a minimum of 200 epochs, a learning rate of 5e-5, and a weight decay of 1e-4. The size of the image and the length of the text embedding were 320 and 77, respectively, according to previous work [23]. The models trained for the third part of the experiments mentioned in the model

definition section were trained for a minimum of 50 epochs with early stopping by validation loss. The Adam optimizer with an initial learning rate of 5e-5, a weight decay of 1e-4, cosine annealing learning rate scheduler, and binary cross-entropy with logits (BCEwithLogitLoss) was used as a loss function, which is a suitable option for multi-label classification. All experiments were implemented using PyTorch and conducted on Nvidia A100 GPUs with 80GB of memory.

## Evaluation metrics

In the first part of the experiments, we evaluated generated images and the synthetic dataset using several complementary metrics. Frechet Inception Distance (FID) and Kernel Inception Distance (KID) capture the distributional similarity between generated and real images, providing a standard measure of generative quality. Structural Similarity Index (SSIM) [24] was employed to assess perceptual and structural fidelity, while Peak Signal-to-Noise Ratio (PSNR) [25] quantified reconstruction quality relative to noise. In addition, training time (hours) was recorded to evaluate the computational efficiency of each approach.

For the second part, model classification performance was assessed with the area under the curve (AUC) and expected calibration error (ECE). To analyze model focus and attention quality, we adopted widely used medical image analysis metrics. Dice Similarity Coefficient (DSC) [26] and Intersection over Union (IoU) [27] measured the overlap between predicted attention regions and reference areas, indicating localization accuracy. SSIM quantified the perceptual similarity of attention maps, while PSNR measured their signal quality relative to noise. Higher values across these metrics reflect better attention and focus quality.

The disparity for all evaluation metrics was computed using the respective formulas following the work of [28].

$$\text{disparity}_{i,\text{sex}} = \underbrace{\left|\text{Perf}_{i,\text{female}} - \text{Perf}_{i,\text{male}}\right|}_{\text{Absolute performance gap between sexes}} \tag{4}$$

$$\text{disparity}_{i,\text{race or age}} = \underbrace{\sum_{j \in \text{subgroup}} \left|\text{Perf}_{i,j} - \text{Med}(\text{Perf}_{i,\text{all}})\right|}_{\text{Sum of absolute deviations from median performance across subgroups}} \tag{5}$$

## Statistical analysis

We used 250 bootstrap samples to obtain a distribution of the AUC and reported 95% CIs. For each bootstrap iteration, we sampled n images with replacement from the test set of n images. To compare the difference in AUC between the proposed model and baseline across all subgroups, we conducted a paired t-test. We obtained statistically significant results with a P-value less than 0.05. To analyze the bias within each dataset, we first employed logistic regression to analyze the association between demographic information (sex, race, and age). These were used as predictors and compared with reference groups such as male vs female, Black vs White, and other races versus Black or White, and individuals older than 71 years versus individuals younger than 71 years old. For the current study, we excluded other races, such as American Indian or Alaska Native, due to low sample size.

## Code and Dataset availability

All codes, the synthetic dataset, and the model weights discussed in this article are publicly available for experimentation and reproducibility via this link:

https://github.com/dawoodrehman44/Stable-Diffusion-with-LoRA

## Results

### CXR image generation

We generated a variety of chest X-rays having a size of 512 × 512, with simple and complex prompts for all the diseases included in the training data, such as *"Chest X-ray of a patient showing Pneumonia,"* and *"Chest X-ray of 62-year-old white male patient showing cardiomegaly and edema"* to generate CXRs with multiple diseases. The combination of prompts and image generation gives us the advantage of control over including and removing the information according to specific situations, as needed. Generated images of the proposed model are shown in Fig 2. We evaluated the quality of synthetic CXRs and the robustness of the proposed model, we further investigated the setup of the target modules while keeping the LoRA ranks the same at the first stage through an ablation study, as shown in Table 1. We observed a significant shift in image quality by increasing target modules. We achieved an optimal FID score of (29.7), KID (0.021), SSIM (0.781), and PSNR (23.8) in setup C. We neglect setups A, B, and D due to small or no improvement in the performance. Similarly, we also performed an ablation study for LoRA ranks configuration while keeping the target modules constant at this stage. Using low ranks, such as 2 and 8, resulted in the worst FID, KID, SSIM, and PSNR scores, while ranks 4 and 16 were optimal parameters for the quality of CXR, as shown in S1 Table. Ranks 16 and 32 slightly improved the quality of the image, but significantly increased the training time by approximately three times. With extensive experiments and observations, we learned that increasing the LoRA rank does not always produce a high-quality image; it could also lead to overfitting. We also performed an experiment related to the t-SNE visualization of latent embedding, indicating potential demographic shortcuts. We observed that embeddings from the synthetic data are more overlapping, suggesting reduced demographic bias and encourages the model to focus on pathology-relevant features rather than demographic cues, as plotted in S1 Fig.

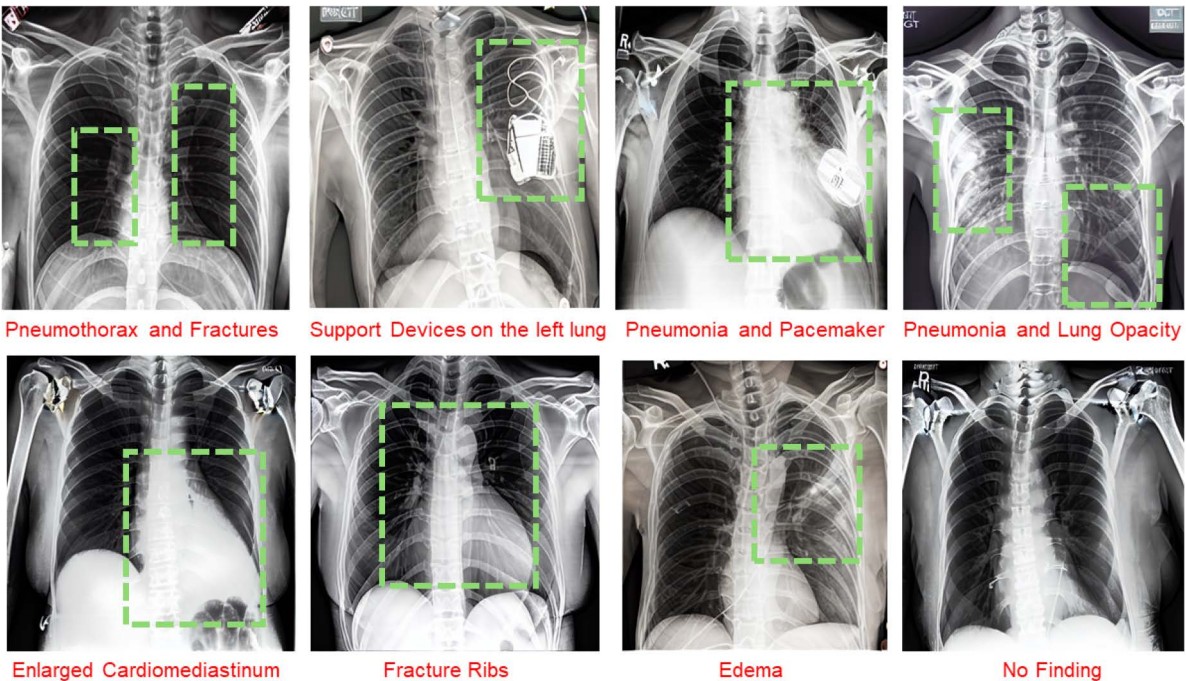

**Fig 2. Examples of synthetic images.** The model generates single and multi-label high-quality disease images for a variety of prompts with medical support devices and specific diseases; the regions are marked in green dashes.

**Table 1. Ablation on Target Modules for LoRA Injection. All configurations use LoRA with rank $r=4$ and scaling factor $\alpha = 16$. Each value represents the mean ± standard deviation over three runs. Lower FID/KID and higher SSIM/PSNR indicate better image quality.**

| Setup | Target Modules | FID ↓ | KID ↓ | SSIM ↑ | PSNR (dB) ↑ | Time (hrs) ↓ |
|---|---|---|---|---|---|---|
| A | `to_q` | 41.3±1.9 | 0.032±0.002 | 0.712±0.008 | 21.4±0.3 | 0.44 |
| B | `to_q, to_k` | 37.5±1.6 | 0.028±0.001 | 0.735±0.007 | 22.1±0.4 | 0.58 |
| C | `to_q, to_k, to_v` | **29.7±1.2** | **0.021±0.001** | **0.781±0.006** | **23.8±0.2** | **1.20** |
| D | `to_q, to_k, to_v, fc1` | 28.3±1.4 | 0.023±0.001 | 0.774±0.005 | 23.2±0.3 | 3.53 |

Table notes: Performance metrics vary depending on the UNet attention submodules targeted for LoRA adaptation. Setup C offers the best quality-efficiency balance. Variability estimates demonstrate stable gains across independent runs.

## Disease classification

Table 2 summarizes the classification results of several state-of-the-art (SOTA) models evaluated under three training scenarios: (i) real data only, (ii) real + synthetic data, and (iii) synthetic data only. Across all models and datasets, incorporating synthetic chest X-rays consistently improved performance in terms of both AUC and calibration (ECE). For instance, in the *real + synthetic* setting, models such as CXR-CLIP (ResNet50) and CheXAgent achieved mean AUC

**Table 2. Classification comparison of models trained on real versus synthetic data. Bold letters indicate improved performance. C, M, CX14, and S represent CheXpert, MIMIC-CXR, ChestX-ray14, and Synthetic datasets, simultaneously. ResNet50 and Swin-T are the CLIP-CXR encoders used for each model.**

| Model | Pre-trained Dataset | CheXpert AUC | CheXpert ECE | MIMIC-CXR AUC | MIMIC-CXR ECE | ChestX-ray14 AUC | ChestX-ray14 ECE | Synthetic AUC | Synthetic ECE |
|---|---|---|---|---|---|---|---|---|---|
| CXR-CLIP$_{Res50}$ [22] | C, M, CX14 | 0.818 | 0.041 | 0.813 | **0.029** | 0.792 | 0.031 | 0.783 | 0.033 |
| CXR-CLIP$_{Res50}$ | C, M, CX14, S | **0.841** | **0.036** | 0.821 | 0.031 | **0.858** | **0.018** | **0.902** | **0.012** |
| CXR-CLIP$_{Res50}$ | S | 0.832 | 0.039 | **0.824** | 0.033 | 0.815 | 0.022 | **0.872** | **0.010** |
| CXR-CLIP$_{Swin-T}$ [22] | C, M, CX14 | 0.813 | 0.037 | 0.781 | 0.025 | 0.788 | 0.036 | 0.811 | 0.021 |
| CXR-CLIP$_{Swin-T}$ | C, M, CX14, S | 0.839 | **0.030** | **0.840** | **0.021** | **0.831** | **0.026** | **0.892** | **0.011** |
| CXR-CLIP$_{Swin-T}$ | S | **0.840** | 0.033 | 0.830 | 0.024 | 0.819 | 0.029 | **0.879** | **0.009** |
| MedCLIP-Xray [23] | C, M, CX14 | 0.801 | 0.044 | 0.792 | 0.038 | 0.774 | 0.041 | 0.789 | 0.030 |
| MedCLIP-Xray | C, M, CX14, S | **0.829** | **0.035** | **0.824** | **0.029** | **0.819** | **0.027** | **0.881** | **0.013** |
| MedCLIP-Xray | S | 0.817 | 0.037 | 0.812 | 0.031 | 0.801 | 0.030 | **0.865** | **0.015** |
| CheXNet [29] | C, M, CX14 | 0.784 | 0.048 | 0.776 | 0.041 | 0.762 | 0.046 | 0.771 | 0.034 |
| CheXNet | C, M, CX14, S | **0.814** | **0.038** | **0.809** | **0.032** | **0.802** | **0.028** | **0.854** | **0.016** |
| CheXNet | S | 0.803 | 0.041 | 0.796 | 0.034 | 0.787 | 0.031 | **0.841** | **0.018** |
| CheXAgent [30] | C, M, CX14 | 0.824 | 0.042 | 0.815 | 0.035 | 0.802 | 0.040 | 0.811 | 0.028 |
| CheXAgent | C, M, CX14, S | **0.851** | **0.033** | **0.844** | **0.026** | **0.832** | **0.023** | **0.896** | **0.012** |
| CheXAgent | S | 0.838 | 0.036 | 0.829 | 0.028 | 0.818 | 0.025 | **0.878** | **0.014** |
| Zhang et al - Fair [4] | C, M, CX14 | 0.806 | 0.046 | 0.798 | 0.039 | 0.783 | 0.043 | 0.795 | 0.032 |
| Zhang et al – Fair | C, M, CX14, S | **0.834** | **0.037** | **0.827** | **0.030** | **0.814** | **0.026** | **0.872** | **0.015** |
| Zhang et al – Fair | S | 0.822 | 0.039 | 0.814 | 0.032 | 0.801 | 0.028 | **0.858** | **0.017** |
| Lin et al – SCL [31] | C, M, CX14 | 0.811 | 0.043 | 0.804 | 0.037 | 0.789 | 0.039 | 0.802 | 0.029 |
| Lin et al - SCL | C, M, CX14, S | **0.842** | **0.034** | **0.836** | **0.027** | **0.824** | **0.024** | **0.889** | **0.013** |
| Lin et al – SCL | S | 0.829 | 0.036 | 0.821 | 0.029 | 0.809 | 0.026 | **0.874** | **0.015** |

Table notes: Synthetic data helps improve classification performance across multiple datasets and encoder architectures.

gains of approximately 2–4% compared to their real-only counterparts, with noticeable reductions in ECE (typically from around 0.04 to below 0.03). The improvements are particularly pronounced on the Synthetic and Chest X-ray14 test sets, where AUCs often exceed 0.85–0.90, indicating strong generalization and consistency of the generated samples with the real distribution. Interestingly, even when trained solely on synthetic data, most models, such as CXR-CLIP (Swin-T) and MedCLIP-Xray, maintained competitive performance, often matching or surpassing real-only training (AUCs in the range of 0.82–0.88). This highlights the realism and diagnostic utility of the synthetic dataset. Among all evaluated models, CXR-CLIP and CheXAgent variants exhibit the most consistent performance across datasets, achieving the highest AUCs (up to 0.90) and lowest calibration errors (as low as 0.01–0.02). Overall, these results demonstrate that synthetic data not only complements real datasets but also independently supports high-quality model training. The consistent improvements across architectures (ResNet and Swin-T) and datasets underscore the strong label alignment, domain fidelity, and diagnostic validity of the generated images (Table 3).

## Model visualization

We applied GradCAM, GradCAM++, and saliency maps to examine the models' focus on specific diseases. The focus of the baseline trained on real data was distracted by shortcuts. For the enlarged cardiomediastinum in the first row of Fig 3, we can see that the baseline model focuses more on the upper and left shoulder to recognize demographic information, and the focus of the model also shifts toward the pacemaker. However, our proposed approach has improved the model's focus significantly towards disease-specific areas. More examples are illustrated in (S2 Fig, S3 Fig, S4 Fig, S5 Fig). S2 Table summarizes the quantitative evaluation of model attention and focus quality using Dice Similarity Coefficient (DSC), Intersection over Union (IoU), Structural Similarity Index (SSIM), and Peak Signal-to-Noise Ratio (PSNR). Higher values indicate more accurate and faithful attention maps. Models trained on the synthetic dataset consistently outperformed those trained on real data. For example, CXR-CLIP$_{Res50}$ achieved DSC 0.88 and IoU 0.78 (vs. 0.85 and 0.75 for real), with

**Table 3. Fairness evaluation: AUC scores for demographic classification. Bold values indicate models trained with synthetic balancing (Res50, Swin-T) show improved fairness.**

| Demographics | Res50 | Swin-T | MedCLIP | CheXNet | CheXAgent | Zhang et al | Lin et al |
|---|---|---|---|---|---|---|---|
| **Gender** | | | | | | | |
| Male | 0.903 | 0.907 | 0.960 | 0.951 | 0.792 | 0.779 | 0.802 |
| Female | 0.892 | 0.895 | 0.932 | 0.920 | 0.771 | 0.768 | 0.786 |
| Average AUC | 0.898 | 0.901 | 0.946 | 0.936 | 0.782 | 0.774 | 0.794 |
| Difference | **0.011** | **0.012** | 0.028 | 0.031 | 0.021 | 0.011 | 0.016 |
| **Race** | | | | | | | |
| Asian | 0.895 | 0.900 | 0.958 | 0.964 | 0.805 | 0.783 | 0.821 |
| Black | 0.888 | 0.892 | 0.925 | 0.915 | 0.787 | 0.777 | 0.798 |
| White | 0.902 | 0.905 | 0.904 | 0.893 | 0.799 | 0.814 | 0.806 |
| Average AUC | 0.895 | 0.899 | 0.929 | 0.924 | 0.797 | 0.791 | 0.808 |
| Difference | **0.014** | **0.013** | 0.054 | 0.071 | 0.018 | 0.037 | 0.023 |
| **Age** | | | | | | | |
| 0–30 | 0.888 | 0.891 | 0.924 | 0.916 | 0.782 | 0.772 | 0.801 |
| 31–50 | 0.896 | 0.898 | 0.902 | 0.895 | 0.796 | 0.807 | 0.803 |
| 51–70 | 0.905 | 0.908 | 0.959 | 0.948 | 0.798 | 0.781 | 0.818 |
| 71+ | 0.900 | 0.902 | 0.938 | 0.929 | 0.786 | 0.776 | 0.796 |
| Average AUC | 0.897 | 0.900 | 0.931 | 0.922 | 0.791 | 0.784 | 0.804 |
| Difference | **0.017** | **0.017** | 0.035 | 0.033 | 0.016 | 0.031 | 0.017 |

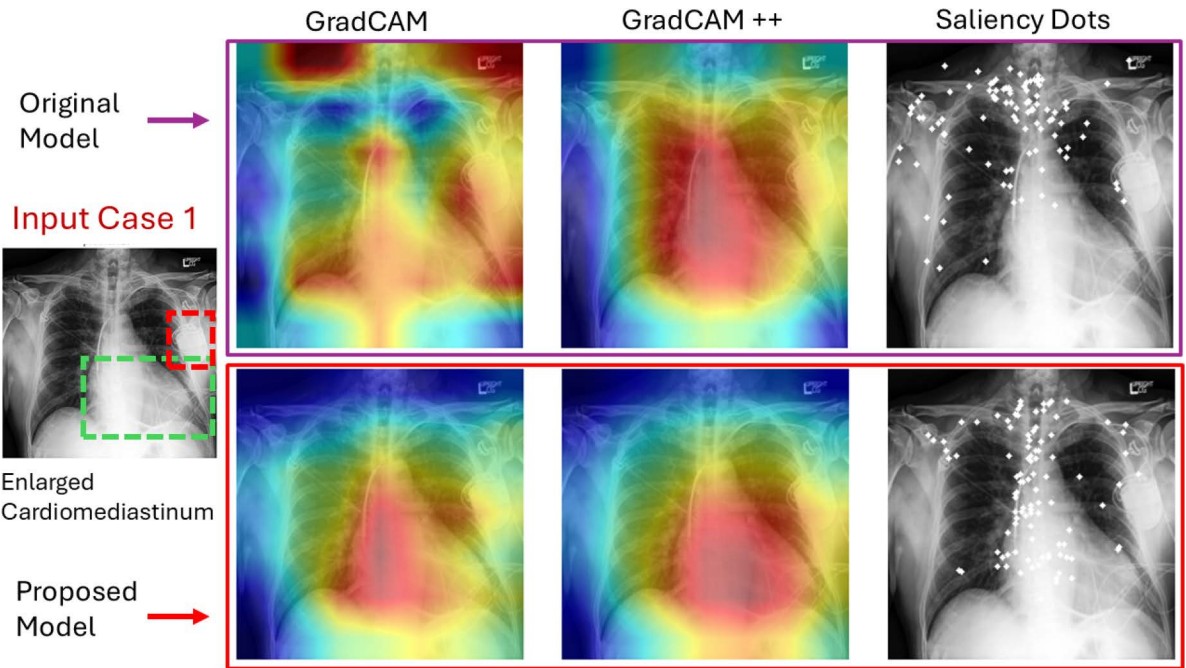

**Fig 3. Model's focus observation comparison.** The top row presents the focus of the baseline model; the lower row shows the proposed model's performance. The green square dashes visualize the disease area, while the red square shows support devices.

SSIM 0.92 and PSNR 36.5 dB (vs. 0.90 and 35.2 dB). Similar improvements were observed for CXR-CLIP$_{Swin-T}$ models. These results indicate that synthetic data enhances the model's focus on disease-relevant regions, producing more accurate and visually consistent attention maps, thereby improving interpretability and confidence in detecting disease patterns. It means that the proposed approach could not only reduce demographic disparity but also boost model focus and lead to a better disease classification performance.

## Fairness analysis

We evaluated model performance across demographic groups, including gender, race, and age. Table 2 presents AUC scores for each group and highlights the differences between synthetic (S) and real data training. Models trained on synthetic data consistently show reduced disparities across all demographic groups. For example, CXR-CLIP Res50 achieved an AUC difference of 0.011 for gender, 0.014 for race, and 0.017 for age, indicating minimal bias. Similarly, CXR-CLIP Swin-T exhibited small differences of 0.012, 0.013, and 0.017, respectively. In contrast, models trained on real datasets showed larger disparities, such as 0.030–0.075 for race and 0.011–0.036 for age. These results demonstrate that synthetic data training effectively reduces demographic bias, making model predictions more equitable across gender, race, and age groups, suggesting that demographic information is less encoded in the learned features. The results are calculated through the disparity equation for each demographic group. The performance of the fairness analysis of the subgroup can be further observed with the evidence in Fig 4. We can see that in each group, the disparity has been reduced for most of the disease cases, and the models trained on synthetic data with the proposed method identify the difference between each demographic group, such as Black, White, and Asian, which is a significant step towards fairness presented in Table 4. For further fairness verification, we used different complementary fairness metrics. This evaluation demonstrates that our approach has enabled the models trained on synthetic data to reduce fairness between groups. Next, we further investigated the effect of increasing the proportion of synthetic data on model performance and fairness (S3 Table). As the fraction of synthetic data grew, model AUC steadily improved for

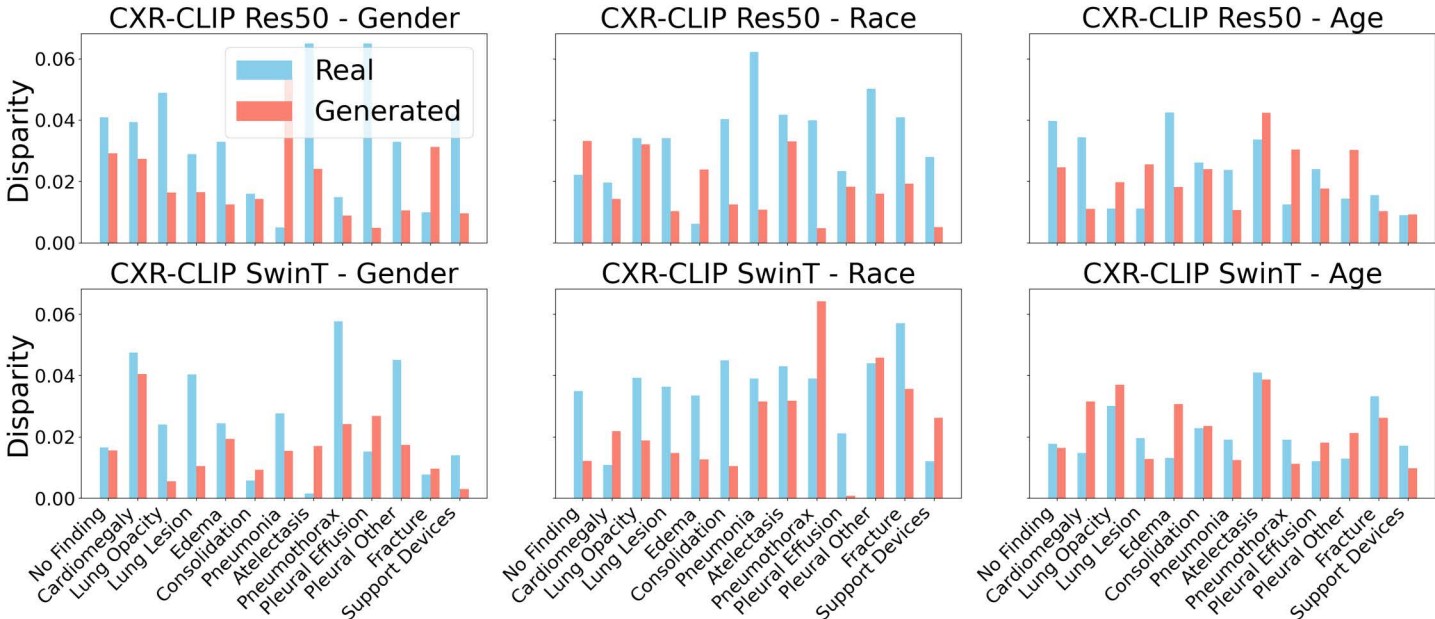

**Fig 4. Subgroup disparity analysis.** This figure shows disparity metrics across gender, race, and age subgroups. Lower values indicate improved fairness. Synthetic data training reduces demographic bias in model features.

**Table 4. Complementary fairness metrics analysis.** Values show fairness gaps across ResNet-50 and Swin-Tiny. Metrics include Equalized Odds (EOD), Demographic Parity (DPD), PPV, and TPR across gender, race, and age. Lower values indicate smaller demographic disparities.

**Res-50**

| Demographic | Equalized Odds | Demographic Parity | PPV | TPR |
|---|---|---|---|---|
| Gender | 0.021 | 0.018 | 0.012 | 0.010 |
| Race | 0.037 | 0.025 | 0.015 | 0.019 |
| Age | 0.042 | 0.031 | 0.017 | 0.020 |

**Swin-T**

| Demographic | Equalized Odds | Demographic Parity | PPV | TPR |
|---|---|---|---|---|
| Gender | 0.025 | 0.021 | 0.015 | 0.013 |
| Race | 0.041 | 0.029 | 0.018 | 0.022 |
| Age | 0.048 | 0.034 | 0.021 | 0.024 |

EOD (Equalized Odds) = $\max\big(|\text{TPR}_a - \text{TPR}_b|, |\text{FPR}_a - \text{FPR}_b|\big)$;

DPD (Demographic Parity) = $\big|P(\hat{Y} = 1 | A = a) - P(\hat{Y} = 1 | A = b)\big|$;

PPV = $\big|\text{PPV}_a - \text{PPV}_b\big|$; TPR = $\big|\text{TPR}_a - \text{TPR}_b\big|$.

Lower values indicate better fairness.

Res50, from 0.911 at 0% to 0.932 at 100% synthetic data. Simultaneously, demographic disparities decreased: the gender gap reduced from 0.030 to 0.003, the race gap from 0.062 to 0.010, and the age gap from 0.058 to 0.008. Swin-T showed similar improvements, demonstrating that incorporating more synthetic data consistently enhances both predictive performance and fairness across gender, race, and age groups.

The results given from all experiments indicate statistically significant ($p < 0.05$) improvements in all datasets and metrics, and highlight that the classification performance of the model has SOTA performance as well as outperforms previous

methods on different occasions. Similarly, we achieved low disparity, which is a key problem that previously reduced the model's detection capability and led to bias. Using generative AI, such as diffusion models, specifically stable diffusion with LoRA, enables us to generate images with proper prompt control and include the necessary information equally. In real datasets, it is hard to exclude information such as demographic information and support devices. Even when data are explicitly selected for disease labels, they could still contain information about demographics that is not required and, in return, affect the performance of a deep learning model. We did not completely remove the demographic information from the image; indeed, the proposed approach gives us an advantage of including and removing information according to a specific task and situation. Using this, we improved model classification and detection capability and reduced disparity, leveraging the robustness of generative AI.

## Discussion

Our results reveal a notable finding: models trained solely on synthetic chest X-rays from our LoRA-adapted Stable Diffusion framework outperform those trained on real data, while mixed training (real + synthetic) achieves the best accuracy. This hierarchy, mixed > synthetic-only > real-only, indicates that the diffusion model not only captures key diagnostic features but also introduces beneficial regularization through anatomically diverse, plausible variations. Interpretability analyses (Grad-CAM, Grad-CAM++, saliency maps) confirm that models trained on synthetic data focus more on clinically relevant regions rather than spurious correlations. Moreover, increasing the proportion of synthetic images substantially reduces demographic performance disparities, showing improved fairness across sex, race, and age. Together, these results position synthetic data generation as a principled strategy to enhance accuracy, interpretability, and fairness in medical AI.

Two complementary mechanisms likely explain these gains. First, distributional balancing: the synthetic corpus was designed to equalize disease labels and enrich demographic attributes, which counteracts real-world sampling biases. The subgroup bar plots in Fig 4. visualize reduced disparity when models are exposed to synthetic data. Second, regularization via plausible diversity: LoRA-tuned Stable Diffusion synthesizes anatomically coherent variants that reduce short-cut learning. The t-SNE analysis (S1 Fig) shows more overlap in latent embeddings for synthetic data, suggesting less demographic encoding and a shift toward pathology-relevant features.

From a deployment perspective, three aspects are salient. (i) Better calibration (lower ECE in Table 2.) can support safer thresholding for triage tools. (ii) Improved focus quality (Fig 3.) increases trust by aligning saliency with radiologically meaningful regions. (iii) Parameter-efficient tuning lowers the cost of refreshing models as demographic distributions evolve, which is crucial for maintaining fairness over time. Together, these findings suggest a pragmatic route to equitable and auditable CXR AI: pair diffusion-based balancing with modern vision-language encoders and measure fairness with multiple metrics.

Despite strong aggregate results, prompt sensitivity and limited spatial reasoning remain open issues for the image generator. The fine-tuned stable diffusion model struggled with consistency when generating CXRs from detailed prompts, such as *"Chest X-ray of a patient with support devices such as a pacemaker on both lungs," "Chest X-ray of a patient with support devices such as a pacemaker on the base of lungs,"* while it was working fine for *" Chest X-ray of a patient with support devices," "medical support devices,"* and *"medical support devices such as pacemaker."* Similarly, it also struggled for diseases, for example, *"Chest X-ray of a patient showing severe cardiomegaly and marked enlargement of the cardiomediastinal silhouette" and "Chest X-ray of a patient showing pronounced enlargement of the heart and mediastinum,"* however, it works perfectly for our adopted prompts, such as *"Chest X-ray of a patient showing enlarged cardiomediastinum" and "Chest X-ray of a 60-year-old white male patient showing enlarged cardiomediastinum ".* This suggests that the model struggles with spatial reasoning and fine-grained localization due to the inaccessibility of information in the training data. S6 Fig highlights information related to the challenges presented in the supporting information.

## Conclusion

This study presents a novel framework that integrates a LoRA-tuned Stable Diffusion model into chest radiograph classification to address demographic bias and improve model robustness. By generating realistic and demographically diverse synthetic CXRs, the approach enhances dataset balance, mitigates representation gaps across sex, race, and age subgroups, and reduces bias in downstream predictive tasks. The framework not only improves overall classification performance but also demonstrates more accurate localization of disease-relevant regions, supporting interpretability and clinical relevance. By leveraging the parameter-efficient LoRA fine-tuning strategy, the model achieves these improvements without the need for extensive computational resources, making it feasible for scalable deployment. This work highlights the potential of combining generative AI with efficient fine-tuning techniques to create domain-adaptable, equitable, and clinically meaningful AI tools.

## Supporting information

**S1 File. Implementation Details and Algorithm.** LoRA Ranks Ablation Study. Effect of varying LoRA rank and target modules on generation quality and efficiency.
(PDF)

**S1 Table. LoRA Ranks Ablation Study.** Effect of varying LoRA rank and target modules on generation quality and efficiency.
(PDF)

**S2 Table. Quantitative evaluation of model attention and focus quality using standard metrics.** Metrics include Dice Similarity Coefficient (DSC), Intersection over Union (IoU), Structural Similarity Index (SSIM), and Peak Signal-to-Noise Ratio (PSNR). Higher values indicate better focus quality and attention map fidelity.
(PDF)

**S3 Table. Importance of Synthetic Data.** Effect of synthetic data proportion on performance and fairness. Performance (AUC) and demographic fairness (average difference across groups) are shown for Res50 and Swin-T models.
(PDF)

**S1 Fig. t-SNE embedding visualisation comparison between real versus synthetic data.**
(PDF)

**S2 Fig. No Finding: Model Focus Observation Case Study 2.**
(PDF)

**S3 Fig. Pneumonia and Enlarged Cardiomediastinum: Model Focus Observation Case Study 3.**
(PDF)

**S4 Fig. Cardiomegaly: Model Focus Observation Case Study 4.**
(PDF)

**S5 Fig. Edema: Model Focus Observation Case Study 5.**
(PDF)

**S6 Fig. Challenges of the proposed framework with specific prompts.**
(PDF)

## Acknowledgments

The authors would like to thank the members of the research community for their insightful discussions and open-source contributions that enabled this work. We gratefully acknowledge the use of publicly available datasets, including CheXpert, MIMIC-CXR, and ChestX-ray14, as well as the open-source Stable Diffusion and CLIP frameworks, which served as the foundation for our experiments. This work was supported by the Ministry of Science and Technology, Taiwan (114–2628-E-007–008-MY4) and U of I-UAAT Joint Research Project (113M7054).

## Author contributions

**Conceptualization:** Dawood Rehman, Po-Chih Kuo.

**Data curation:** Dawood Rehman.

**Formal analysis:** Dawood Rehman.

**Investigation:** Dawood Rehman, Chi-Chun Lee.

**Methodology:** Dawood Rehman, Huan-Yu Chen, Po-Chih Kuo.

**Project administration:** Chi-Chun Lee, Po-Chih Kuo.

**Resources:** Chi-Chun Lee, Po-Chih Kuo.

**Supervision:** Chi-Chun Lee, Po-Chih Kuo.

**Validation:** Huan-Yu Chen, Chi-Chun Lee, Po-Chih Kuo.

**Visualization:** Dawood Rehman, Po-Chih Kuo.

**Writing – original draft:** Dawood Rehman, Po-Chih Kuo.

**Writing – review & editing:** Dawood Rehman, Huan-Yu Chen, Chi-Chun Lee, Natalia Vilor-Tejedor, Shreya Upadhyay, Po-Chih Kuo.

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
